# Comparison of the Micronaut-AM System and the EUCAST Broth Microdilution Reference Method for MIC Determination of Four Antifungals against *Aspergillus fumigatus*

**DOI:** 10.3390/jof9070721

**Published:** 2023-06-30

**Authors:** Nikolett Gyurtane Szabo, Valentin Joste, Sandrine Houzé, Eric Dannaoui, Christine Bonnal

**Affiliations:** 1Laboratoire de Parasitologie-Mycologie, AP-HP, CHU Bichat-Claude-Bernard, 75018 Paris, France; valentin.joste@aphp.fr (V.J.); sandrine.houze@aphp.fr (S.H.); 2MERIT, IRD, Université Paris Cité, 75006 Paris, France; 3Unité de Parasitologie-Mycologie, Service de Microbiologie, AP-HP, Hôpital Européen Georges Pompidou, 75015 Paris, France; eric.dannaoui@aphp.fr; 4DYNAMYC 7380, Faculté de Santé, Université Paris-Est Créteil (UPEC), 94010 Créteil, France; 5Faculté de Médecine, Université Paris Cité, 75006 Paris, France

**Keywords:** antifungal susceptibility testing, Micronaut-AM, EUCAST, *Aspergillus fumigatus*, *cyp51A* gene, azole resistance

## Abstract

The Antifungal Susceptibility Testing method of the European Committee on Antimicrobial Susceptibility Testing (EUCAST-AFST) is a reference technique for the determination of the Minimum Inhibitory Concentration (MIC) of antifungals for *Aspergillus fumigatus*. However, it is time-consuming and requires expertise. Micronaut-AM (M-AM) is a fast, simple, time-saving, and ready-to-use new colorimetric method using an indicator (resazurin) to facilitate the visual reading. The aim of this retrospective study was to evaluate the performance of the M-AM system and compare it with the EUCAST broth microdilution reference method to determine the susceptibility of 77 *A. fumigatus* clinical strains to amphotericin B, itraconazole, voriconazole, and posaconazole. Overall, the essential agreements within ±2 dilutions were 100%, 62%, 58%, and 30% and the categorical agreements were 100%, 97%, 91%, and 87% for amphotericin B, itraconazole, voriconazole, and posaconazole, respectively. No categorical discrepancy was found for amphotericin B, but several categorical discordances were observed with azole antifungals. However, only 2 of the 16 azole-resistant strains confirmed by the *cyp51A* sequencing would have been misclassified by M-AM. The use of M-AM is probably suitable for the determination of the MICs of amphotericin B, but further evaluations are needed to confirm its usefulness for the determination of the MICs of azoles for *A. fumigatus*.

## 1. Introduction

*Aspergillus fumigatus* is the most frequent etiological agent responsible for human aspergillosis [1]. Triazole antifungal agents (voriconazole, itraconazole, isavuconazole, and posaconazole) are recommended as primary treatment for *Aspergillus*-related infections [2]. Azoles act by inhibiting the activity of cytochrome P450 14α-sterol demethylase (Cyp51A and Cyp51B), thereby blocking the demethylation of C-14 of lanosterol, resulting in the blockage of conversion of lanosterol to ergosterol, which is an essential cell membrane component of filamentous fungi. It results in the accumulation of 14-alpha methyl sterols in the fungal cell and at the end of synthesis of ergosterol, which damages cell transport and fungal membrane structure [3]. The lanosterol 14α-demethylase enzyme is a member of the cytochrome P450 family and is encoded by the *cyp51A* gene [4]. Azole resistance is a trait that can be acquired during long-term azole treatment in patients (patient route), or through exposure of *A. fumigatus* to azole fungicides in the environment (environmental route) [3,5,6]. Azole resistance acquired during a long-term treatment is mostly due to single point mutations in the *cyp51A* gene resulting in amino-acid substitutions within the Cyp51A protein (mostly G54, G138, P216, F219, M220, Y431, and G448) [7]. These mutations may directly block the entry or modify the binding site of the drug, thereby reducing affinity of drug–enzyme interaction. The environmental exposure to azole results in a mutation point located in the *cyp51A* gene associated with the insertion of tandem repeats (TR) in the promoter region (mainly TR_34_/L98H and TR_46_/Y121F/T289A) [7,8]. The presence of the TR repeat has been shown to drive overexpression of *cyp51A* by increasing the binding activity of the sterol regulatory element binding protein, resulting in increased tolerance of azoles [7,8]. The average incidence rate of azole resistance is estimated to be 0.6–4.2% worldwide [8]. One major consequence of azole resistance is the failure of treatment [9,10]. Clinical studies have shown that two-thirds of patients with azole-resistant infections had no previous history of azole therapy. Moreover, high mortality rates of between 50% and 100% are reported in azole-resistant invasive aspergillosis. In this context, azole monotherapy should be avoided and therefore few therapeutic options are still available to treat infections with azole-resistant *A. fumigatus*. Liposomal amphotericin B or a combination of voriconazole and an echinocandin is recommended [10].

Early detection of resistance is of paramount importance. It can be performed by different methods, either broth microdilution methods or concentration gradient strip methods. The European Committee on Antimicrobial Susceptibility Testing (EUCAST) and the Clinical and Laboratory Standards Institute (CLSI) broth microdilution methods are the reference techniques for in vitro antifungal susceptibility of filamentous fungi [11,12,13]. Reference techniques require the preparation of dilutions of each antifungal. It is time-consuming and the visual reading of MICs needs expertise. Unlike the reference method, the concentration gradient strip (GCS) technique is one of the most routinely used techniques in laboratories because it is easier to perform [11]. However, it also has some limitations, mainly due to the difficulty to read the results and the absence of clinical breakpoints for some antifungal drugs following the CLSI’s standards.

If the azole-resistance is proven with susceptibility testing, it must be confirmed by molecular analysis. Sanger sequencing is widely used to detect mutations associated with azole resistance in the *cyp51A* gene. However, this analysis is difficult to perform routinely, with a delayed response that is not compatible with the treatment of the patient.

Recently, a ready-to-use colorimetric method, the Micronaut-AM Antifungal Agents MIC^®^ (MERLIN Diagnostika GmbH, Bornheim, Germany) (M-AM), has been commercialized to determine antifungal susceptibility of *Candida* spp. and *Cryptococcus* spp. The ready-to-use plate contains 9 dehydrated antifungals in up to 11 concentrations (anidulafungin: 0.002–8 mg/L, micafungin: 0.002–8 mg/L, caspofungin: 0.002–8 mg/L, fluconazole: 0.002–128 mg/L, posaconazole: 0.0078–8 mg/L, voriconazole: 0.0078–8 mg/L, itraconazole: 0.031–4 mg/L, amphotericin B: 0.031–16 mg/L, and 5-flucytosine: 0.0625–32 mg/L). Its MICRONAUT-RPMI 1640 medium (including 3-(N-morpholino) propanesulfonic acid (MOPS) and glucose) is used to improve the growth of fungi. The AST indicator (resazurin) added to the test medium facilitates the visual reading [14]. It is possible to read visually the MICs of antifungals or automatically use the Micronaut Skan device and the Micronaut software [15]. Therefore, it is a fast, simple, time-saving method, adapted for the routine use, but it has not yet been validated for the determination of antifungal susceptibility of *A. fumigatus*.

The aim of this study was to compare the M-AM plate and the EUCAST reference method to evaluate the susceptibility of *A. fumigatus* isolates to amphotericin B, itraconazole, voriconazole, and posaconazole.

## 2. Materials and Methods

### 2.1. Fungal Isolates

Seventy-seven *A. fumigatus* clinical strains were included in this retrospective study. Isolates were obtained from the samples of 70 hospitalized patients of 2 French hospitals (Hospital Européen Georges Pompidou AP-HP and Hospital Bichat-Claude-Bernard AP-HP) and stored at −80 °C. After thawing, the strains were cultured on Sabouraud dextrose agar and incubated at 35 °C for 2–5 days to prepare fresh, mature cultures. All of the isolates were *A. fumigatus* sensu stricto, identified by matrix-assisted laser desorption ionization time-of-flight mass spectrometry (MALDI-TOF MS). For all these strains, the azole susceptibility had been tested using the GCS method before storing at −80 °C. The strains with an elevated MIC to azoles (itraconazole MIC ≥ 2 mg/L, voriconazole MIC ≥ 2 mg/L, posaconazole MIC ≥ 0.5 mg/L) were considered as resistant strains (*n* = 16). For these strains, the *cyp51A* gene was secondarily sequenced using the Sanger technique to search for mutations potentially implicated in the azole resistance. The remaining 61 *A. fumigatus* strains had no elevated MIC to azoles using the GCS method. They were classified as sensible strains. *A. fumigatus* ATCC 204305 was used as a quality control isolate [13,16].

### 2.2. The Micronaut-AM Method

This susceptibility testing was performed by following the M-AM guideline [15]. Briefly, 0.5 McFarland conidia suspension was prepared in NaCl 0.9%, which is equivalent to ~1–5 × 10^6^ cfu/mL. The suspension was then 1:1000 diluted into 11.5 mL of MICRONAUT-RPMI-1640 Medium (containing 4% of glucose and MOPS) supplemented with 100 µL of AST indicator (Rezasurin) [15]. Then, 100 µL of working solution containing 1–5 × 10^3^ cfu/mL was distributed into each well with a multi-channel pipette. The plates were incubated at 35 °C. They were read visually under normal laboratory lighting. Rezasurin is an oxidation-reduction indicator. With sufficient fungal metabolism and growth, the oxidized blue indicator switches to the reduced state, which has a pink color [15]. After 24 h of incubation, the positive growth control well was examined. When the color of the growth control had changed from blue to pink, the MIC values for the antifungal agents were read out. It is possible that the produced resorufin (pink) underwent further reduction to dihydroresorufin (white) due to intensive fungal growth. In case the color of the growth control was still blue or faintly purple, the test plates were re-incubated and re-examined after 48 h. The MIC was read according to the recommendations of the manufacturers as the lowest concentration of each antifungal indicated by color changed from blue to pink or white [15]. Only results regarding four antifungals (amphotericin B, itraconazole, voriconazole, and posaconazole) were considered in this study. The determination of the MICs of echinocandins was not possible because it requires the reading of Minimum Effective Concentrations and is therefore not adapted to the visual reading.

### 2.3. The EUCAST Method

This susceptibility testing was performed by following the EUCAST E.DEF 9.3.2 guideline [13]. The antifungal drugs used (Sigma Aldrich^®,^ Saint Louis, MO, USA) were amphotericin B, itraconazole, voriconazole, and posaconazole. The range of the concentrations tested was 0.03–16 mg/L for amphotericin B and voriconazole, 0.016–8 mg/L for itraconazole and posaconazole [13]. Two-fold serial dilutions of the antifungal drugs were prepared according to ISO recommendation and then the antifungals were diluted in RPMI 1640 (Gibco™ Roswell Park Memorial Institute, Thermo Fisher Scientific, Asniere-sur-Seine, France) supplemented with 2% of glucose and MOPS [13]. One hundred microliters of conidia suspension was added to each well of the plate containing RPMI/MOPS medium (resulting in a final concentration of 1–2.5 × 10^5^ cfu/mL) except for the last column, 12 (used as a negative growth control). The plates were incubated at 35 °C and were read visually with a complete inhibition endpoint after 48 h of incubation [13].

For each strain, the two methods (M-AM and EUCAST) were performed from the same culture at the same time.

### 2.4. The cyp51A Gene Sequencing

For the 16 isolates identified as being phenotypically resistant to azole antifungal with the GCS method, the entire *cyp51A* gene and its promoter region were sequenced. DNA was extracted from the suspension of 0.5 McFarland of the strain using a nucleic acid extraction system (EMAG^®^, bioMerieux, Paris, France). PCR conditions and DNA sequencing were prepared by the method described by Mortensen [17]. Eight PCR primers were used to sequence the *cyp51A* gene and its promoter region [17]. The products of PCR were sequenced on the ABI 3130 (Thermo Fisher Scientific, Asnière-sur-Seine, France). The sequences obtained were compared to the *cyp51A* sequence of the *A. fumigatus* strain AF338659.1 [18].

### 2.5. Data Analysis

Essential agreement (EA) and categorical agreement (CA) were used to investigate the performance of the two methods [11]. The MICs of strains obtained by the M-AM system were compared with the MICs read visually by the EUCAST method to evaluate the EA and CA. EA is defined as the percentage of agreement within ±1 or ±2 log_2_ dilutions between the two methods. CA is the percentage of isolates interpreted with the two methods in the same category (susceptible, ATU: Area of Technical Uncertainty or resistant) according to the clinical breakpoint of the EUCAST [19]. The geometric mean (GM) of the MICs values was calculated using Microsoft Excel 2010 software for each antifungal. Discordances were classified as minor (isolate was classified susceptible or resistant by one method but ATU with the other), major (strain categorized susceptible by the EUCAST versus resistant by the M-AM), or very major discrepancy (strain categorized resistant by the EUCAST versus susceptible by the M-AM).

## 3. Results

For the 77 isolates of *A. fumigatus*, the GM of the MICs of amphotericin B was 0.77 mg/L with the EUCAST technique and 0.98 mg/L with the M-AM. The GMs of the MICs with the EUCAST method were 0.46 mg/L, 0.40 mg/L, and 0.20 mg/L for itraconazole, voriconazole, and posaconazole, but were lower (0.13 mg/L, 0.07 mg/L, and 0.03 mg/L) with the M-AM method for these three antifungals (Table 1). The comparison of the MICs/distributions for the four antifungals obtained by the two methods is presented in Figure 1.

For the *A. fumigatus* ATCC 204305 control strain, lower MICs of azole antifungals were also observed with the M-AM technique (itraconazole: 0.03 mg/L, voriconazole: 0.03 mg/L, posaconazole: 0.008 mg/L) than with the EUCAST technique (itraconazole: 0.125 mg/L, voriconazole: 0.25 mg/L, posaconazole: 0.06 mg/L). The MICs of amphotericin B for this control strain were identical (1 mg/L) with both methods.

The MICs measured for amphotericin B were within ±2 dilutions in both methods for all isolates, which resulted in a CA and EA of 100%. On the other hand, the M-AM system showed a high CA (97% for itraconazole, 91% for voriconazole, and 87% for posaconazole) but a low EA (62% for itraconazole, 58% for voriconazole, and 30% for posaconazole) for azole antifungals because the MICs measured were always lower with the M-AM method than with the EUCAST technique (Table 1).

For 61 strains, no resistance to azoles and amphotericin B was measured by the GCS method. These strains were also susceptible to amphotericin B, itraconazole, voriconazole, and posaconazole with the two methods (M-AM and EUCAST). Appendix A, shows the comparison of MICs (mg/L) and categories for the 61 azole-susceptible isolates.

On the other hand, we found categorical discrepancies for the 16 azole-resistant strains, so that we analyzed them in more detail. For these strains, the *cyp51A* gene analysis revealed that eight strains had a TR_34_/L98H alteration and one had a TR_46_/Y121F/T289A mutation (Table 2). Single point mutations in the *cyp51A* gene resulting in amino-acid substitutions within the Cyp51A protein were detected in the other strains (G54E in two strains; G54W in one strain; Y121H in one strain; P216L in one strain; F46Y, M172V, N248T, D255E, E427K in one strain). In the isolate N°1, no mutation was found within the *cyp51A* gene, so it was considered to be wild type (WT) (Table 2). It should be noted that Y121H amino-acid substitution has not been previously reported in the literature.

The azole susceptibility of the 16 azole-resistant strains was globally categorized as susceptible, ATU, or resistant to the azole antifungals in order to evaluate the ability of M-AM to detect phenotypically azole-resistant strains harboring the mutations of the *cyp51A* gene. Therefore, an isolate was considered as resistant or ATU if it was categorized resistant or ATU to at least one azole drug (itraconazole, voriconazole, or posaconazole). Fourteen strains were identified to be phenotypically azole-resistant and two strains to be ATU with the EUCAST method. Eleven strains were identified to be phenotypically azole-resistant, three strains to be ATU, and two strains to be S with the M-AM method (Table 2). For voriconazole, nine categorical discrepancies were noted: two very major (R with the EUCAST method and S with the M-AM method) and seven minor (five ATU with the EUCAST method and S with the M-AM method in five strains, and two R with the EUCAST method and ATU with the M-AM method). These discrepancies were observed in eight azole-resistant isolates with the mutation TR_34_/L98H and one isolate with the mutation Y121H. It is worth noting that six of the strains with a mutation in the *cyp51A* were S with both techniques.

Furthermore, we found no major or very major errors for itraconazole, but five minor errors (three R with the EUCAST method and ATU with the M-AM method, two ATU with the EUCAST method and S with the M-AM method). Only 1/16 strain (N°10 TR46/Y121F/T289A) was categorized S with both methods.

The most important discordances were found with posaconazole. Twelve categorical discrepancies were found: seven very major (R with the EUCAST method and S with the M-AM method) mainly with the strains with the TR34/L98H mutation (5/7) and five minor (two R with the EUCAST method and ATU with the M-AM method, and two ATU with the EUCAST method, and S with the M-AM method). Moreover, two strains (N°12 F46Y, M172V, N248T, D255E, and N°14 G54R) were susceptible with both techniques (Table 2).

## 4. Discussion

The objective of this study was to evaluate the performance of the M-AM system by comparing it with the EUCAST broth microdilution reference method to determine the susceptibility of *A. fumigatus* to amphotericin B, itraconazole, voriconazole, and posaconazole.

Globally, the MICs of itraconazole, voriconazole, and posaconazole using M-AM were always lower than with EUCAST. The GM of MICs obtained by M-AM were approximately three-fold dilution lower for voriconazole and posaconazole, and two-fold dilution lower for itraconazole than those observed using the EUCAST method. For amphotericin B, the MICs obtained by the two methods were almost identical, within ±1 log_2_ dilution. As a result, the overall CA between the two methods was found to be high for amphotericin B (100%), for itraconazole (97%), for voriconazole (91%) and to be moderate for posaconazole (87%). A study published by Nuh et al. evaluated the M-AM system by comparing it with the reference CLSI technique for 78 clinical isolates of the Aspergillus species [20]. They found higher EA and CA for azole antifungals, which were 90% and 97% for voriconazole, and 87% and 99% for itraconazole, respectively [20]. The EA of amphotericin B was identical to those observed by Nuh et al., but higher CA (100% vs. 96%) was found with the M-AM method in our study [20]. It is possible that the absence of amphotericin B-resistant isolates could have led to the higher CA in our study.

However, the M-AM assay allowed the identification of azole resistance for all resistant isolates except for two (N°6 TR_34_/L98H and N°14 G54E). These two strains had intermediate MIC values with EUCAST, suggesting a weak expression of the resistance in vitro. Furthermore, there were three isolates (N°1 WT, N°4 TR_34_/L98H, N°8 TR_34_/L98H) categorized to be ATU with M-AM but resistant with EUCAST. The MIC values of itraconazole of these strains (2 mg/L) with the M-AM method suggest that they could be azole-resistant isolates, even though they are susceptible to posaconazole and voriconazole. In this case, it is necessary to perform additional tests such as the *cyp51A* sequencing or other antifungal susceptibility testing in order not to miss an isolate potentially resistant to azoles.

The EUCAST method is very difficult to use as a routine technique. Therefore, we were very interested to know whether M-AM could be used for the susceptibility screening of *A. fumigatus* in the routine clinical laboratory. The detection of the resistance to posaconazole and voriconazole was not good, particularly for the strains with the TR_34_/L98H mutation. Our results are in accordance with those published by Nuh et al., who reported very major discrepancies in two azole-resistant *A. fumigatus* strains (interpreted resistant to voriconazole by CLSI versus susceptible by M-AM) [20].

The reasons for the discrepancies of the azole MICs between the two methods (M-AM and EUCAST) are not clear but there are a few hypotheses. For the reference method for filamentous fungi published by the Clinical Laboratory Standard Institute (CLSI), it is well known that the size of inoculum, the type of growth medium, the time of incubation, and the inoculum preparation method can influence MIC values [21]. Several studies showed the importance of inoculum preparation in the haemocytometer for accurate and reproducible preparation independent of the color and size of the conidia. To limit the influence of the inoculum preparation, the two methods (M-AM and EUCAST) were performed from the same culture at the same time. However, the final working inoculum was 1–5 × 10^3^ cfu/mL with the M-AM method and 100-fold more concentrated with the EUCAST technique 2–5 × 10^5^ cfu/mL [13,15]. Moreover, the EUCAST method includes glucose in the RPMI broth for in vitro susceptibility testing of antifungals against Aspergillus spp. because it enhances growth and facilitates the determination of endpoints [21]. Therefore, the difference in the concentration of glucose in the growing medium of EUCAST (2%) of M-AM (4%) may play a role in the discordance of the azole MICs. Moreover, in our study, we did not use a Micronaut Skan device and Micronaut software, which would have probably resulted in a more accurate reading of the MICs. However, as the correlation between the two techniques is excellent for amphotericin B, it is unlikely that the differences are related to the inoculum or to the glucose concentration in the growing medium. Further studies are required to validate these hypotheses.

On a broader scale, the culture conditions and the reading of the different antifungal susceptibility techniques (CLSI, EUCAST, GCS, Sensititre YeastOne^®^ (Thermo Fisher Scientific, Asniere-sur-Seine, France), etc.) could influence the MIC values [22,23,24]. Dellière et al. compared the results obtained by the EUCAST method and the GCS method in strains with the mutation TR_34_/L98H. They found nine categorical discrepancies (eight minor errors and one strain interpreted resistant by the EUCAST versus susceptible by the GCS) [25]. Another study comparing a colorimetric method (Sensititre YeastOne^®^ (Thermo Fisher Scientific, Asniere-sur-Seine, France)) with CLSI found also very major discrepancies (interpreted resistant to voriconazole by CLSI versus susceptible by Sensititre YeastOne^®^ (Thermo Fisher Scientific, Asnière-sur-Seine, France) in three *A. fumigatus* with the mutation TR_34_/L98H [26]. These results are consistent with our own and emphasize the need for additional studies on defining the clinical breakpoints according to the method used.

In the routine clinical laboratory, it is important to use an in vitro antifungal susceptibility assay that can detect the isolates resistant to at least one azole drug. These strains could then be further studied with other techniques (determination of MICs by the reference EUCAST method, the *cyp51A* sequencing) to confirm the resistance to azoles. In our study, none of the susceptible strains and only two of sixteen azole-resistant strains (categorized S instead of ATU) would have been wrongly categorized with M-AM. No very major discrepancy was detected by using the global azole categorization. Moreover, for the first isolate (N°1), no mutation was found in the *cyp51A* gene whereas the two methods detected an elevated MIC for the itraconazole (8 mg/L with the EUCAST method and 2 mg/L for the M-AM method), suggesting that another mechanism of azole resistance was involved. Therefore, it is advised to analyze the results according to the interpretation of the three azoles (itraconazole, voriconazole, and posaconazole) in order to identify a resistant strain. Itraconazole seems to be the most suitable drug for the resistance screening because itraconazole resistance was detected by M-AM for most strains (10 of 16 strains). The supplementing of M-AM plate with isavuconazole might increase the possibility of detecting azole resistance.

Furthermore, we used the clinical breakpoints of EUCAST suggested by the M-AM manufacturer’s protocol to categorize the isolates. If it is confirmed that MICs measured by this method are always lower than those obtained with EUCAST, these breakpoints should probably be revised on the basis of the *cyp51A* sequencing and the MICs obtained with the M-AM system. We were not able to do this in our study because the number of strains tested is not large enough.

In conclusion, the use of the M-AM technique is probably suitable for the determination of the MICs of amphotericin B, although it must be confirmed by the evaluation of amphotericin B-resistant strains. However, our findings show that this new colorimetric method presents limits for the evaluation of the azole resistance of *A. fumigatus.* Therefore, more studies are needed to confirm the possibility of using M-AM for the screening of the susceptibility of *A. fumigatus* to azole antifungals in routine laboratories.

## Figures and Tables

**Figure 1 jof-09-00721-f001:**
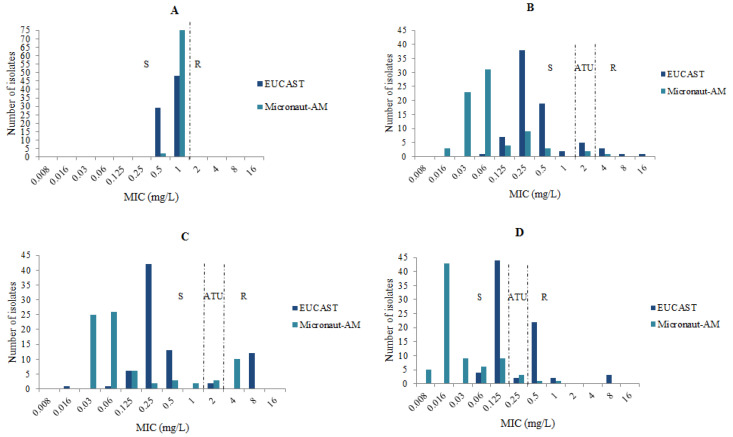
MICs’ distributions of the antifungals tested for 77 *A. fumigatus* isolates with the EUCAST and the M-AM techniques. (**A**): amphotericin B, (**B**): voriconazole, (**C**): itraconazole, (**D**): posaconazole, S: susceptible, ATU: Area of Technical Uncertainty, R: resistant.

**Table 1 jof-09-00721-t001:** The geometric mean (GM) of Minimum Inhibitory Concentrations (MICs) values, MICs’ range, the essential (EA) and categorical agreement (CA) between the EUCAST (E) and the M-AM (M) for 77 *A. fumigatus* strains.

The GM, MIC Range, the EA within ±2 Dilutions and CA for 77 *A. fumigatus* Strains						
	GM (mg/L)		MICs Range (mg/L)		EA	CA
	E	M	E	M		
Amphotericin B	0.77	0.98	0.5–1	0.5–1	100%	100%
Itraconazole	0.46	0.13	0.016–8	0.03–4	62%	97%
Voriconazole	0.40	0.07	0.06–16	0.016–4	58%	91%
Posaconazole	0.20	0.03	0.06–8	0.008–1	30%	87%

**Table 2 jof-09-00721-t002:** Comparison of MICs (mg/L) and categories for 16 azole-resistant isolates.

N° Isolate	Mutation(s) in *cyp51A*	Voriconazole			Itraconazole			Posaconazole			Global Azole Categorization
		MIC (mg/L)		C	MIC (mg/L)		C	MIC (mg/L)		C	
		E	M	E/M	E	M	E/M	E	M	E/M	E/M
1	WT	2	0.25	ATU/S	8	2	R/ATU	0.5	0.06	R/S	R/ATU
2	TR_34_/L98H	4	0.25	R/S	8	4	R/R	0.5	0.125	R/S	R/R
3	TR_34_/L98H	2	0.25	ATU/S	8	4	R/R	0.5	0.125	R/S	R/R
4	TR_34_/L98H	4	2	R/ATU	8	2	R/ATU	0.5	0.125	R/S	R/ATU
5	TR_34_/L98H	2	0.25	ATU/S	8	4	R/R	0.5	0.25	R/ATU	R/R
6	TR_34_/L98H	1	0.25	S/S	2	0.125	ATU/S	0.25	0.0625	ATU/S	ATU/S
7	TR_34_/L98H	2	0.5	ATU/S	8	4	R/R	1	0.125	R/S	R/R
8	TR_34_/L98H	2	0.25	ATU/S	8	2	R/ATU	0.5	0.125	R/S	R/ATU
9	TR_34_/L98H	4	0.5	R/S	8	4	R/R	0.25	0.125	ATU/S	R/R
10	TR_46_/Y121F/ T289A	16	4	R/R	0.5	0.0625	S/S	0.5	0.0625	R/S	R/R
11	Y121H	8	2	R/ATU	8	4	R/R	8	0.5	R/R	R/R
12	F46Y, M172V, N248T, D255E, E427K	0.5	0.25	S/S	8	4	R/R	0.125	0.125	S/S	R/R
13	G54E	0.5	0.03125	S/S	8	4	R/R	8	0.25	R/ATU	R/R
14	G54E	0.5	0.03125	S/S	2	1	ATU/S	0.125	0.0625	S/S	ATU/S
15	G54W	0.5	0.25	S/S	8	4	R/R	8	1	R/R	R/R
16	P216L	0.5	0.0625	S/S	8	4	R/R	1	0.25	R/ATU	R/R

MIC: Minimum Inhibitory Concentration, C: Category, E: EUCAST, M: Micronaut-AM, S: susceptible, ATU: Area of Technical Uncertainty, R: resistant, WT: wild-type, TR_34_/L98H: tandem repeat of 34 base pairs with L98H substitution, TR_46_/Y121F/T289A: tandem repeat of 46 base pairs with Y121F and T289A substitution.

## Data Availability

All the data are present in the manuscript.

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
