# Peer review of "Comparison of the Micronaut-AM System and the EUCAST Broth Microdilution Reference Method for MIC Determination of Four Antifungals against *Aspergillus fumigatus"

_jof, 2023, doi:10.3390/jof9070721_

Round 1
Reviewer 1 Report
The paper by Szabo and colleagues aims to make easier the detection of azole resistance in routine settings.
The manuscript is overall well written, data and results are clear, with the exception of some minor criticisms:
- Lines 44-47, the paper could be improved adding the azole targets
- Line 117, 11.5 mL de MICRONAUT...?
- Line 140, with 2% de glucose...?
- Line 156, resistant to azole antifungal using which of the two methods? Furthermore, authors included also isolates that resulted S/ATU, not only the actually resistants.
- Lines 177-178, the meaning is difficult to understand. Please, rephrase.
- Line 190 and the rest of the paper, A. fumigatus should be italicized
- Could figure 1 be bigger? It is hard to read.
- Line 217, add the number of the strains (16)
- Line 225, azole-resistant or ATU
- Table 2, the caption does not include the mutations. Moreover, please add the reference genome
There are only few typos and sentences difficult to understand, is it correct the "so that" at line 215?
Reviewer 2 Report
In this paper, the authors aimed to compare a commercially available method for the antifungal susceptibility testing of Aspergillus fumigatus sensu stricto to the reference EUCAST method. This is reasonable as the manufacturer of the commercial panel recommends the use of EUCAST breakpoints to interpret the results of the test. For this purpose, the authors test in parallel a collection of 77 (clinical?) isolates using the commercial panel and the EUCAST standard.
Whereas, in general, the results are interesting to microbiologists working in routine labs without access to the standard, the manuscript could be substantially improved by keeping the focus on the stated aim.
The introduction is too long and makes a review of subjects that, despite being important, are not particularly relevant for the purpose of the study: comparing a commercial panel to eucast and assessing the proficiency of the commercial panel to provide accurate results as compared to the standard in order to correctly identify wild type and non-wild type patterns of susceptibility. Thus, my suggestion is to focus the introduction on the following points:
-What is the natural susceptibility pattern of A fumigatus, and what is the extent and clinical meaning of triazole resistance in this species (in short).
-Why susceptibility to A fumigatus has to be studies in routine labs, what are the possible alternatives to do so, and what problems routine labs have to face: lack of accessibility to standard methods, and commercial tests not reproducing exactly the standard so recommended standard breakpoints cannot be directly extrapolated to interpret their results. Need for studies assessing the performance of commercial methods as compared to the standard.
In the methods section, some minor modifications can be introduced:
-Please state the origin of the isolates (clinical or environmental?).
-Criteria for the selection of these particular 77 isolates (consecutive?). It is not easy to understand that some isolates were previously screened whereas other were not.
-Lines 107-114 contain a description of the Micronaut section that can be deleted, moved to the introduction, or included in the discussion along with a mention to the sensititre method.
-Lines 124-125 should state clearly whether the micronaut was read visually or with the help of a reader (it is mentioned in lines 176-178, but it would be more appropriate to insert the statement here)
-Lines 130-151 are a description of the EUCAST methodology, which is freely available at the eucast website, so it would be more appropriate to shorten this paragraph and to insert a reference or a link.
-I would like to recommend shorten the description of the cyp51A sequencing, just mentioning the differences between the method described in reference 22 and the one used by the authors, if any.
-Definitions of categorical and essential agreement are standardized. Please, insert an appropriate reference.
The results section could be slightly re-arranged in order to present data more clearly.
-I would suggest to include in table 1 a general summary of the results: MIC range, GM, CMIc range, EA and CA for each of the two methods, keeping the text to comment on the important finding that micronaut underestimates de MICs of the triazoles as compared to eucast (which is nicely shown in fig 1).
-The precise nature of non-wild type of the isolates is not the aim of the study, buy so it is to prove whether the abnormal phenotypic profile of non-wild type isolates (altered cyp51A sequence) identified with the help of eucast is also detected by using micronaut. In this line, point 3.2 can be deleted from the text, table 2 can be summarized (put together the results of TR34strains, the 2 G54 islates, and the, individually, the results of the other mutations) following the format indicated for table 1). Table 2 in its current form seems more appropriate to be included as supplementary material. Moreover, I would suggest to elaborate the results of all the strains included in the same way to be included as suppl. mat.
-The content of lines 230-243 are not results. Those statements will fit better in the discussion.
The discussion can be improved by focusing it on the aim of the study. It seems appropriate to discuss whether commercial systems reproduce or not the results of clsi or eucast, as well as the findings of other authors. In most cases, what has been found is that the criteria released by eucast or clsi do not allow to categorize correctly the strains when they are tested by commercial methods, and that specific breakpoints would be better. An interesting contribution thar could be done by the authors is a suggestion of interpretation based on their own findings: probably, categorical agreement would be better if they are reinterpreted on the basis of the cyp51 sequencing plus the mics obtained with the micronaut system. Obviously, the number of strains tested is not large enough to set categorical recommendations, but it is nice to suggest a modification in the interpretation criteria.
Lines 278-293 seem to discuss a subject that is out of the scope of the paper (again, the aim of the work is not seeking the impact of mutations on the phenotype. The aim is to assess whether the micronaut is able to identify aberrant patterns as well as eucast). I recommend to delete them.
The reasons for discrepancies between methods are pretty well discussed, although I think they miss an important point: differences between standards and commercial methods can be attributed not only to the size of the inoculum, but also to the amount of glucose used to supplement RPMI, and I think this should be indicated in lines 306-321.
The conclusion is ok, but I would suggest the inclusion of a short commentary on the need for method-specific criteria to improve the yield of the method.
Language review by a native speaker would also be advisable to amend minor grammatical mistakes.
Author Response
Please see the attachement.
